# Shiga-Toxin Producing *Escherichia Coli* in Brazil: A Systematic Review

**DOI:** 10.3390/microorganisms7050137

**Published:** 2019-05-16

**Authors:** Vinicius Silva Castro, Eduardo Eustáquio de Souza Figueiredo, Kim Stanford, Tim McAllister, Carlos Adam Conte-Junior

**Affiliations:** 1Institute of Chemistry, Universidade Federal do Rio de Janeiro, 21941-909 Rio de Janeiro, Brazil; viniciuscastro06@gmail.com (V.S.C.); carlosconte@id.uff.br (C.A.C.-J.); 2Agronomy and Animal Science College, Universidade Federal de Mato Grosso, 78060-900 Cuiabá-Mato Grosso, Brazil; 3Nutrition College, Universidade Federal de Mato Grosso, 78060-900 Cuiabá-Mato Grosso, Brazil; 4Department of Food Technology, Faculdade de Veterinária, Universidade Federal Fluminense, 24230-340 Rio de Janeiro, Brazil; 5Alberta Agriculture and Forestry, #100-5401 1st Ave. S, Lethbridge, AB T1J 4V6, Canada; Kim.Stanford@gov.ab.ca; 6Agriculture and Agri-Food Canada, Lethbridge Research and Development Centre, 5403 1st Avenue South, Lethbridge, AB T1J 4B1, Canada; Tim.McAllister@agr.gc.ca; 7National Institute of Health Quality Control, Fundação Oswaldo Cruz, Rio de Janeiro, 21040-900 Rio de Janeiro, Brazil

**Keywords:** STEC, food microbiology, food-borne diseases, VTEC, EHEC, shiga-toxigenic *Escherichia coli*, bloody diarrhea

## Abstract

Shiga-toxin producing *E. coli* (STEC) can cause serious illnesses, including hemorrhagic colitis and hemolytic uremic syndrome. This is the first systematic review of STEC in Brazil, and will report the main serogroups detected in animals, food products and foodborne diseases. Data were obtained from online databases accessed in January 2019. Papers were selected from each database using the Mesh term entries. Although no human disease outbreaks in Brazil related to STEC has been reported, the presence of several serogroups such as O157 and O111 has been verified in animals, food, and humans. Moreover, other serogroups monitored by international federal agencies and involved in outbreak cases worldwide were detected, and other unusual strains were involved in some isolated individual cases of foodborne disease, such as serotype O118:H16 and serogroup O165. The epidemiological data presented herein indicates the presence of several pathogenic serogroups, including O157:H7, O26, O103, and O111, which have been linked to disease outbreaks worldwide. As available data are concentrated in the Sao Paulo state and almost completely lacking in outlying regions, epidemiological monitoring in Brazil for STEC needs to be expanded and food safety standards for this pathogen should be aligned to that of the food safety standards of international bodies.

## 1. Introduction

Brazil is one of the largest producers and exporters of food in the world. Animal products, such as beef, poultry, pork, fish, and crops such as corn, soybeans, and rice represent Brazil’s major exports [1]. However, it is a challenge to maintain both high production efficiency and control physical, chemical and microbiological contamination [2]. In addition, the production of different animals and crops may be carried out in the same geographic region and the use of animal manure as a fertilizer may promote the contamination of fruits and vegetables [3].

The main microorganisms involved in food contamination belong to the *Enterobacteriaceae*, with *Escherichia coli* representing a major species. This bacterium represents one of the most extensively studied and was one of the first to be sequenced [4,5]. In addition, *Escherichia coli* is the main bacterium involved in food contamination in Brazil [6]. It possesses seven pathogenic groups, namely enterotoxigenic (ETEC), enteroinvasive (EIEC), enteropathogenic (EPEC) diffusely adherent (DAEC), invasive adherent (AIEC), enteroaggregative (EAEC), and Shiga-toxin producing (STEC) [7].

These pathogenic groups are involved in outbreaks related to food consumption, and STEC in particular are of importance to public health, potentially causing diarrhea, bloody diarrhea (hemorrhagic colitis), hemolytic uremic syndrome (HUS) and renal injury [8,9,10]. STEC strains may produce two immunologically distinct toxins: Shiga-toxin type 1 and 2. In addition, STEC strains may have a pathogenicity island, LEE (Locus of Enterocyte Effacement), which encodes the proteins that include those responsible for induction of attaching-and-effacing lesions [11,12].

In Brazil, there are no reports of outbreaks involving STEC. The hypothesis is that: (i) Disease outbreaks are not being recorded due to lack of a centralized reporting system or (ii) disease outbreaks are not being recognized as there is no surveillance system for STEC. Farm animals have been shown to be carriers and contamination of food as well as sporadic STEC infections in humans have been reported [13,14]. In addition, STEC infections have been reported in several South American countries, including an endemic issue in Argentina, which borders on Brazil. The possible reasons for this contrast are the differences in cattle breeds, surveillance systems, more livestock confinement system or a combination of these factors. Moreover, Brazil’s large cattle populations pose a direct risk, as STEC infections are mainly linked to beef and milk consumption. Furthermore, understanding the relationships among the STEC serogroups in the different Brazilian regions and sources (livestock, food, humans) in recent years is crucial for the future monitoring and control strategies concerning this relevant pathogen [4]. There are some documents that have compiled global STEC data based on sites of health institutions and overview by continent [15,16]. However, the prevalence and distribution of STEC serogroups in Brazil remains unclear. In this context, the aim of the present study was to conduct the first systematic review of *Escherichia coli* STEC with a focus on Brazil and to compare the presence of serogroups detected in food products, animals and humans.

## 2. Materials and Methods

Data were obtained from online databases PubMed, Scielo, Lilacs, Web of Science and Cochrane BVS. The date interval filter was set from January 2000 to December 2018, accessed between 10 September 2018 and 2 January 2019. Papers were selected according to the Prisma guidelines and flow diagram [17]. Therefore, from each database using the Mesh term entries: “Verotoxigenic *Escherichia coli*” OR “Verotoxigenic” OR “STEC” OR “Shiga Toxigenic *E. coli*” OR “Shiga Toxigenic *Escherichia coli*” OR “Shiga Toxin-Producing *Escherichia coli*” OR “VTEC” OR “Vero Cytotoxin-Producing *Escherichia coli*” OR “Verotoxigenic *E. coli*” OR “Verotoxigenic *Escherichia coli*” OR “Verotoxin-Producing *Escherichia coli*” AND “Brazil”. Papers in both English and Portuguese were included in this review. Duplicates were traced and excluded. A total of 161 papers were collected, independent of sample size or culture/detection methods. The full text for each paper was obtained and evaluated individually (Figure 1). Papers were excluded when reporting the characterization of strains isolated from another published article, when assessing decreases in intentional contamination (inactivation methods), and when composed of literature reviews or experimental infection. After the application of these criteria, a total of 80 papers were selected.

## 3. Results

### 3.1. Animals

Data on livestock were evaluated in 35 scientific articles, analyzing the presence of STEC through the collection of feces from healthy animals and those with diarrhea (Table 1). The frequency of STEC was heterogeneous and determination of the presence of STEC in herds presents a challenge. STEC contamination rates in cattle ranged from 17.5% to 71.0%, and relevant serogroups O157:H7, O113:H21 and O111 were detected.

In calves, STEC prevalence rates ranged from 12.0% to 20.9%, and serogroups O26, O103 and O111 were detected. These serogroups are monitored in meat by control agencies, such as the EFSA (European Food Safety Authority) and USDA-FSIS (Food Safety and Inspection Service), which require their absence in meat products. In addition, STEC were isolated from 2.7% to 78.3% of sheep and O103:H2 was detected in sheep (Table 1) and has been associated with human disease cases in Brazil [18]. Other animals were also positive for STEC such as pigs (2.2%) and rabbits (5.1%). In chickens, a study evaluated 110 strains of avian pathogenic *Escherichia coli* (APEC) and noted that *stx*_1_ and *stx*_2_ were present in 30.9% of the strains, indicating possible dispersion of the *stx* genes between STEC and APEC. In addition, *stx*_1_ and *stx*_2_ were detected in 4.7% of poultry litter samples in the south of Brazil (Table 1).

In other species that could possibly act as vectors, STEC rates were 0.7% to 20.4% in wild birds and 15% in stable flies (*Stomoxys calcitrans*). Prevalence rates in dogs ranged between 0 and 48%, but to date there are no reports of STEC in cats in Brazil.

### 3.2. Food

The Shiga toxin-producing contamination in food was evaluated in 23 scientific articles (Table 2). In six studies, the presence of STEC in beef was assessed, with prevalence rates ranged from 0 to 27.5%, the serogroups O157 and the “big six” (O26, O45, O103, O111, O121, and O145) were not isolated. In milk, the presence of *stx*_1_ and *stx*_2_ in *E. coli* ranged from 0 to 31.1%, but isolates were not serotyped in these studies [52,53,54]. However, STEC were detected in 0 to 14% of cheese samples. In addition, the presence of O111, O55, and O157:H7, serogroups frequently linked to food-borne disease outbreaks worldwide [55] were also found in Brazilian cheese.

Contaminated water is increasingly linked to STEC outbreaks associated with fruits and vegetables in Europe [56]. Although STEC contamination is usually related to products of animal-origin, contamination of plant products occurs as a result of cross-contamination [57]. In Brazil, STEC prevalence rates in water ranged from 0.65 to 5.93% with only a single sample testing positive for O157:H7. Water can also be a source of contamination of plant products. For example, in a study with lettuce, 0.76% of samples were contaminated with O157:H7.

### 3.3. Human

The presence of STEC contamination in humans was evaluated in 22 scientific articles (Table 3). In five studies, STEC infection was reported in case reports or in samples collected from patients with hemolytic uremic syndrome, and was associated with O26:H11, O103:H2, O165, O157, O157:H7 and O104:H4 (enteroaggregative group with the acquisition of the *stx* gene). Through whole genome sequencing, the O104:H4 strain in Brazil was found to be similar to a strain isolated from an American citizen diagnosed with hemolytic uremic syndrome (HUS) who had traveled to Germany during the 2011 HUS outbreak [14].

In other studies of diarrhea or healthy subjects, the serogroups predominantly detected were O26:H11, O103:H2, O111 Not typeable (NT), O118:H16, O165:NT and O157:H7 (Figure 2).

Some of these serogroups were reported in animal and food studies in Brazil. For instance, the O111 and O157 serogroups were verified in all three groups. Our hypothesis is that the dispersion of the strains has contaminated all stages of the food chain (pre and post-processing). Moreover, the O103 strains in animals were linked to those found in humans. In contrast, stains O165 and O118:H16 were only detected in human clinical cases.

## 4. Discussion

The most frequent serogroups described in peer-reviewed papers in samples collected in Brazil are represented in Figure 2. As with studies in other countries, O157:H7 was the serotype most frequently linked to human cases and also had the highest occurrence rates in animal, food and humans in Brazil. However, due to the large-scale outbreaks related to this serotype in the USA in 1982 and 1993, much effort has been invested in methods to detect this specific serotype. It can be readily isolated as non-sorbitol fermenting colonies (the main O157 characteristic) and identified using PCR primers designed specifically for the O157 antigen or the flagellum H7. Consequently, a higher prevalence of O157:H7 in Brazil might also be related to its ease of detection.

Serogroup O111 was also reported in animals, food and humans and has been linked to clinical human cases in Brazil [13,66,77] and worldwide [55,97]. O111 is one of six non-O157 serogroups classified by the USDA-FSIS (2013) as foodborne pathogens that should be monitored during meat production. Considering the serogroups listed by the USDA-FSIS, three: O157, O111, O26:H11 and O103:H2 have been associated with foodborne disease in Brazil (Figure 2). However, two other non-O157 strains, O165 and O118:H16 are not part of the USDA-FSIS “big six”, but were linked to cases of diarrhea, hemolytic uremic syndrome or hemorrhagic colitis in Brazil.

Serogroup O165 has been reported in cases of hemolytic uremic syndrome in Japan [98], and Germany [99]. This serotype has also been identified in cattle in the United States [100]. In addition, its genome has recently been made available on the GenBank/NCBI platform [101] for comparisons and genetic investigations, emphasizing the increasing importance of this serogroup in food production and human infection. Serotype O118:H16 has been linked to foodborne diseases in Germany [102], the United States [103], and its genome sequence was included in GenBank/NCBI in 2014 [104]. Detection of other new STEC in future cases of human disease is likely.

Studies evaluating different cattle feeds indicate that different diets may increase the acid resistance of *Escherichia coli* strains, enabling the bacteria to survive during passage through the human stomach [105]. Strain resistance due to diet may be responsible for the selection of some strains of STEC in Brazilian herds. In addition, a recent study by Acquaotta [106] identified a correlation between STEC infections and climatic differences in North America, where more cases of contamination were reported during warm periods as compared to cold periods. This result emphasizes the need for monitoring and control during food production as Brazil’s tropical climate may increase the risk of STEC infections.

Foods showed the greatest diversity among serogroups, followed by animal sources (Figure 2). This diversity may be related to several factors, such as breeding and production heterogeneity, the nature of the animal and food carrier, the innate (animal) or initial (food) microbial load and the applied methodology (antibiotic use for pre-enrichment, colony morphology and analyzed genes). Moreover, serogroups detected in humans display greater homogeneity, perhaps related to toxin subtypes (*stx*_2e_) or other genetic factors including virulence [107,108].

Studies were concentrated mainly in Brazil’s most developed areas and especially in the state of Sao Paulo (Figure 3), with the southern and southeastern regions accounting for 69% of the Brazilian population [6]. However, the majority of animal and grain-production occurs in central Brazil, which is responsible for 34.4% of animal and 42% of grain production [109]. Differences in the number of STEC studies directly reflect national development and demographics. Further studies in the central region would provide a better description of STEC present in livestock and food and would aid in obtaining a more accurate national picture of foodborne STEC risks.

Currently, the main legislation in force in Brazil concerning food production is Resolution 12 of 2001 [110], which applies to coliforms in frozen or fresh meat. This resolution, however, does not call for STEC serogroup analyses in any food products. However, the Ministry of Livestock and Food Supply (MAPA) established an internal standard in 2013, requesting *E. coli* STEC tests for O157 and the “big six” non-O157 in beef destined for export [111]. However, any data about the frequency of STEC isolation has not been released.

The analysis procedure determined by MAPA is based on the methodology used by the USDA FSIS [112], comprising molecular screening followed by cultivation on Rainbow™ agar O157 (Biolog Inc, Hayward, USA) and serology of the positive isolates. The use of this methodology assists in production control and monitoring of the main STEC, followed by subsequent implementation of measures to combat contamination. However, this measure is only for beef, and it is necessary to extend the standard to other food matrices as the present review demonstrated high STEC contamination rates in milk and cheese, as well as water. The STEC causing infection and for which microbiological monitoring and control during production is required show some geographical variation and new STEC continually evolve. Instead of continuing to add new serogroups to a long list of food adulterants, microbiological monitoring could instead aim for the absence of strains that possess Shiga-toxin gene(s) in their genome.

Moreover, the improvement and cost reduction of molecular tools such as whole genomic sequencing (WGS), metagenomics, and others will facilitate the understanding of serotype epidemiology and the dispersion of these strains in neighboring countries and in other continents. Epigenetics advances will also improve the understanding of gene expression and the impact of good animal management practices, as well as possible genome mutations that may influence virulence and antimicrobial resistance profiles [113].

## 5. Conclusions

The epidemiological data presented in this review indicate that O157:H7, O26, O103 and O111 strains, classified as foodborne pathogens and monitored by the USDA-FSIS (USA), U.S. FDA, EFSA (European Food Safety Authority - EU) and MAPA (Ministry of Agriculture, Livestock and Supply - Brazil), are currently circulating in different regions of Brazil. Although no STEC outbreak cases in Brazil have been reported, several animal, food and human studies have indicated the presence of STEC in Brazil and it has been related to several foodborne outbreaks around the world. Thus, improved epidemiological monitoring and food production control is necessary. Novel studies should be financed in regions presenting significant agricultural production, such as Brazil’s central region in order to better assess potential threats and prevent human STEC infections.

## Figures and Tables

**Figure 1 microorganisms-07-00137-f001:**
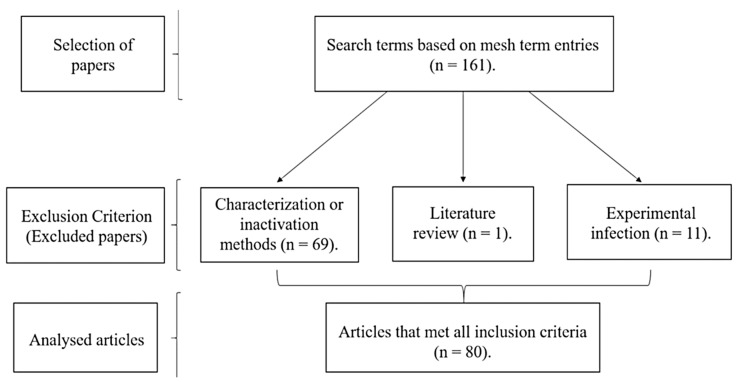
Papers selection and exclusion scheme.

**Figure 2 microorganisms-07-00137-f002:**
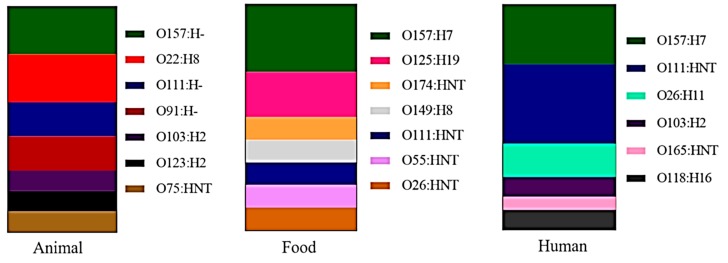
Main serotypes present in each assessed group reported from 2000 to 2018.

**Figure 3 microorganisms-07-00137-f003:**
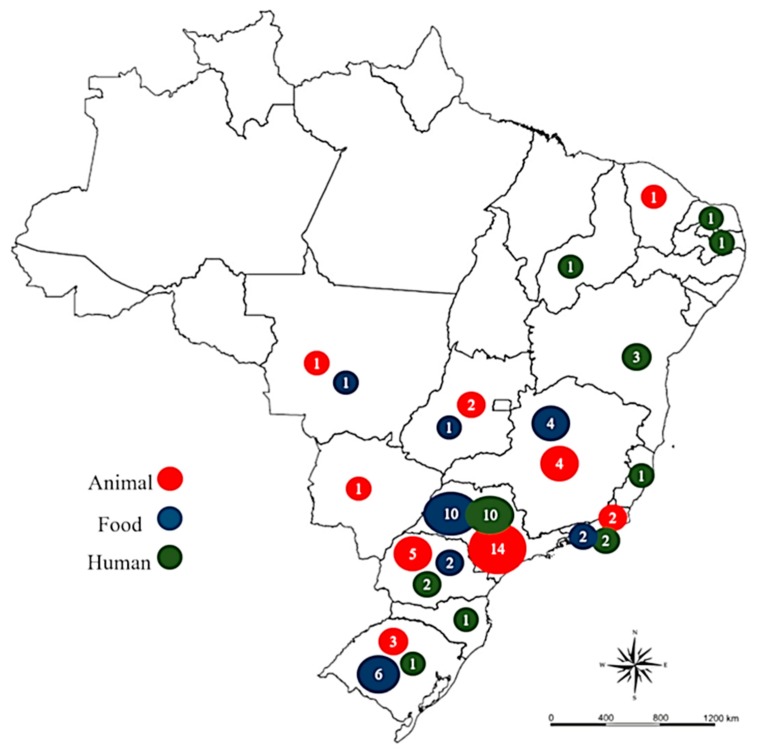
Distribution of STEC assessment studies carried out in Brazil from 2000 to 2018.

**Table 1 microorganisms-07-00137-t001:** Prevalence of Shiga toxin-producing *Escherichia coli* (STEC) isolated in animals reared in Brazil.

Host	State	Number of Samples	Prevalence	Serotype or Genes Amplified	Author
Bovine and Buffaloes	Rio Grande do Sul	243 feces from dairy cattle	49%	O157:H-, O91:H-, O125:H-, O119:H-, O112:H-, O29:H-	[19]
	São Paulo	182 *E. coli* isolated from milk samples	12.08%	*stx*_1_ and *stx*_2_	[20]
	São Paulo	153 fecal samples	25.5%	O113:H21, O157:H7, O111:H-, O22:H8	[13]
	Rio Grande do Sul	243 feces from dairy cattle	48.9%	O157, O157:NM, O91:NM, O112:NM	[21]
	Minas Gerais	100 water buffaloes	37%	O137:H41, O74:H25, O159:H21, O41, O77:H18, O88:H25, O116:H21, O141:H49, O178:H19, O23:H7, O82:H8, O93:H19, O59:H8, O113:H21, O93:H19, O156:H21, O22:H16, O49, O49:H21, O77:H41, O176:H2, O93:H16, O79:H14	[22]
	Minas Gerais	205 healthy beef and dairy cattle, and 106 reared goats	57.5% (goats) 39.2% (beef cattle 17.5% in dairy cattle	O181:H4, O22:H8, O104:H2, O116:H21, O105:H8, O157:H7, O98:H14, O22:H16, O22:H4, O150:H8, O179:H8, O79:H14, O6:H49, O19:H7, O91:H21, O141:H49, O178:H19, O174:H21, O174:H8, O39:H49, O124:H21	[23]
	Paraná	190 healthy cattle	36%	O6:H34, O10:H42, O17:H41, O22:H8, O22:H16, O41:H2, O74:H42, O79:H, O82:H8, O98:H41, O110:H2, O113:H21, O117:H8, O124:H11, O159:H21, O174:H21, O175:H21, O178:H19, O179:H8, O181:H4	[24]
	Goiás	198 rectal swabs	72.73%	*stx*_1_, *stx*_2_, *ehxA*	[25]
	Rio Grande do Sul	108 carcass swabs	20.37%	O157:H7	[26]
Birds	São Paulo	171 fecal samples	19.4%	*stx*_1_ and *stx*_2_	[27]
	São Paulo	516 fecal samples	0.74%	*stx* _2_	[28]
	São Paulo	446 fecal samples	9.8%	O6, O48	[29]
	Minas Gerais	100 fresh fecal samples	2%	*stx*_1_ and *stx*_2_	[30]
	Ceará	Case report	-	*stx* _1_	[31]
	São Paulo	118 pigeons and 38 great egrets	2.5% (pigeons)	*stx* _2f_	[32]
	São Paulo	Case report	-	O137:H6	[33]
Sheep	São Paulo	48 sheep	52.1%	O5:H-, O16:H-, O75:H-, O75:H8, O87:H16, O91:H-, O146:H21, O172:H- OR:H-, ONT:H-, ONT:H16	[34]
	São Paulo	330 feces and 99 carcass samples	2.72% (Feces), 1.01% (Carcass)	O5, O75, and O91	[35]
	Paraná	Case report	-	O103:H2	[36]
	Paraná	130 fecal samples	50%	O76:H19 and O65:H–	[37]
	Goiás	115 *E. coli* strains	78.3%	*stx*_1_, *stx*_2_	[38]
Calves	Mato Grosso do Sul	205 *E. coli* strains	9.75% (*stx*_1_), 6.34% (*stx*_2_)	O26:H, O111:H, O118:H16	[39]
	São Paulo	139 diarrheic and 205 non-diarrheic fecal samples	12.7%	O113:H21, O118:H16, O123:H2, O111:NM, O111:H8	[40]
	São Paulo	264 diarrheic and 282 healthy fecal samples	12%	O7:H7, O7:H10, O48:H7, O111:H19, O123:H2, O132:H51, O173:H(-), O175:H49	[41]
	Paraná	29 diarrheic and 21 healthy fecal samples	101 strains	O1, O3, O7, O8, O17, O23, O78, O144, O146, ONT, O26, O55, O103, O117, O123, O124, O153, O15, O128, O175, O119, O4, O156	[42]
	Minas Gerais	850 fecal samples	20.9%	*stx*_1_, *eae*, *iha*, *toxB*, *ehxA*, *efa-1*, *saa*, *astA*	[43]
Chickens	São Paulo	110 APEC *E. coli* samples	30.90%	*stx*_1_, *stx*_2_	[44]
Pigs	Mato Grosso	74 lumen content samples	2.2%	*stx* _2_	[45]
Dogs	São Paulo	25 feces from diarrheic dogs	48%	O157:H7 and *stx*_1_, *stx*_2_, *eaeA*	[46]
Cats	São Paulo	40 feces samples and 3 urine infection samples	0%	-	[47]
Dogs and Cats	Paraná	50 cat feces and 50 dogs	0%	-	[48]
Rabbits	São Paulo	178 *E. coli* isolates	5.05%	O20:H28, O41:H-, O103:H19, O110:H6, O126:H- O126:H20, O128:H2, O132:H2, O153:H7	[49]
Avian Organic Fertilizers	Paraná	40 fertilizers	4.7%	*stx*_1_, *stx*_2_	[50]
*Stomoxys calcitrans*	Rio de Janeiro	40 *Stomoxys calcitrans* flies	15%	*stx*_1_, *stx*_2_ and *eae*	[51]

**Table 2 microorganisms-07-00137-t002:** Prevalence of Shiga toxin-producing *Escherichia coli* isolated in food produced in Brazil.

Matrix	State	Number of Samples	Prevalence	Serotype or Genes Amplified	Author
Beef	São Paulo	204 bovine carcass swabs	27.5% (rainy season), 17.5% (dry season)	*stx*_1_ and *stx*_2_	[58]
	São Paulo	250 raw ground beef samples	1.6%	O93:H19, O174:HNT	[59]
	São Paulo	91 beef samples	2.1%	*stx* _2_	[60]
	São Paulo	70 raw kibe samples	2.8%	O125:H19, O149:H8	[61]
	São Paulo	552 meat products samples	0%	-	[62]
	Rio Grande do Sul	5 beef jerky samples	0%	-	[63]
	Mato Grosso	80 samples	10%	O83:H19, O26:HNT, O73:H45, O8:H21, O79:H7, O113:H21, O22:H16, O117:H7, O21:H19, O132:H21	[4]
Milk	São Paulo	30 milk samples	3.3%	*stx*_1_, *stx*_2_	[64]
	Minas Gerais Rio Grande do Sul	670 bovine mastitis milk samples	8.6%	*stx*_1_, *stx*_2_	[65]
	Rio Grande do Sul	101 milk samples	31.1%	*stx*_1_, *stx*_2_	[52]
	São Paulo	62 milk samples	0%	-	[53]
	Paraná	87 milk *E. coli* strains	0%	-	[54]
Cheese	Minas Gerais	50 cheese samples	14%	O125, O111, O55, O119	[66]
	Minas Gerais	30 cheese samples	0%	-	[67]
	Goiás	60 cheese samples	6.7%	O157:H7	[68]
	Minas Gerais	147 *E. coli* strains isolated from 38 cheeses	9.5%	*stx* _1_	[69]
Water	São Paulo	133 *E. coli* isolates	0.75%	*stx* _2_	[70]
	Paraná	1850 drinking water samples	0.65%	*ehxA*, *saa*, *lpfA*_O113_, *iha*, *subAB*, *cdtV*	[71]
	Rio de Janeiro	178 *E. coli* isolates	2.8%	*stx* _1_	[72]
	São Paulo	25 water samples	19 isolates	*stx*_2_, *rfbE*_O157:H7_	[73]
	Rio Grande do Sul	219 water samples	5.93%	O157:H7	[74]
Vegetable	Rio Grande do Sul	260 lettuce samples	0.76%	O157:H7	[75]
Shrimp meat	São Paulo	42 chilled shrimp samples	0%	-	[76]

**Table 3 microorganisms-07-00137-t003:** Prevalence of Shiga toxin-producing *Escherichia coli* isolated in cases of foodborne disease in Brazil.

State	Number of Samples	Prevalence	Toxin Type	Serotype	Author
São Paulo	1010 children feces samples	0.3%	*stx* _1_	O111ac	[77]
	3 patient strain samples	Case report	*stx*_1_ and *stx*_2_	O157:H7	[78]
	2607 samples from patients with diarrhea	1.1%	*stx*_1_ and *stx*_2_	O55:H19, O93:H19, O118:H16, O157:H7 O111:HNM, O111:H8, O26:H11	[79]
	1 haemolytic anaemia and 2 faecal with diarrhea samples	Case report	*stx*_1_ and *stx*_2_	O103:H2	[18]
	Feces from 19-month-old children	Case report	*stx* _2_	O165:HNM	[80]
	337 children and 102 HIV adult patients	1.8%	*stx*_1_ and *stx*_2_	O111:HNM, O157:H7, ONT:H2, O103:H2, O118:H16, O77:H18, ONT:H8	[81]
	13 post-diarrheal HUS cases	Case report	*stx*_2_, *stx*_2c_	O26:H11, O157:H7, O165:H-	[82]
	Stool specimens collected from 115 children	0.86%	*stx*_1_ and *stx*_2_	ONT:H19	[83]
	Stools and ileum biopsy of a 51-year-old woman	Case report	*stx* _1_	O104:H4	[14]
	5047 cases of human infection	4.2%	*stx*_1a_, *stx*_1d_, *stx*_2a_, *stx*_2c_, *stx*_2d_, and *stx*_2e_	O8:H19, O24:H4, O26:H11, O71:H8, O91:H14, O100:HNM, O103:HNM, O111:H11, O111:H8, O111:HNM, O118:H16, O123:H2, O123:HNM, O145:HNM, O153:H21, O153:H7, O157:H7, O177:HNM, O178:H19	[84]
Bahia	1233 children feces	1.6%	*stx*_1_, *stx*_2_	O26:H11, O21:H21	[85]
	1233 children feces	1.6%	Not disclosed	0%	[86]
	1207 children feces	0.6%	*stx* _2_	O157:H7, O26:H11, O111:H^-^	[87]
Paraná	306 culture stool samples	0.65%	*stx*_1_, *stx*_2_	O69:H11, O178:H19	[88]
	141 children fecal samples	2.83%	*stx*_1_, *stx*_2_	0%	[89]
Rio de Janeiro	307 children samples	0%	0%	0%	[90]
	Human gastroenteritis caused by *E. coli* O157:NM	Case report	*stx* _2_	O157:NM	[91]
Rio Grande do Norte and Santa Catarina	2 strains	Case report	*stx*_1_ and *stx*_2_	O157	[92]
Rio Grande do Sul	375 children feces samples	0.26%	*stx* _2_	O91:H21	[93]
Piauí	400 children feces samples	0.4%	Not disclosed	O125	[94]
Espírito Santo	560 children feces samples	0.17%	Not disclosed	0%	[95]
Paraíba	580 children feces samples	0%	0%	0%	[96]

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
