# Peer review of "Shiga-Toxin Producing *Escherichia Coli* in Brazil: A Systematic Review"

_microorganisms, 2019, doi:10.3390/microorganisms7050137_

Round 1
Reviewer 1 Report
Manuscript number: 484896
Title: An overview of Shiga-toxin producing Escherichia coli in Brazil: A
systematic review
Author: Castro et al.,
General comments
This is a well written paper. There are a couple of recent publications that should be mentioned in the introduction as they are topical (see list below). The publications are not specific to Brazil do include the region. I think the authors need to be clear what this review is providing that is not in the others.
Majowicz et al, 2014Global Incidence of Human Shiga Toxin–Producing
Escherichia coli Infections and Deaths: A Systematic Review and Knowledge Synthesis. Foodborne Pathogens and Disease.
Torres et al., 2018. Recent Advances in Shiga Toxin-Producing Escherichia coli Research in Latin America. Microorganisms.
WHO report, 2018. Escherichia coli (STEC) and food: attribution, characterization,
and monitoring (https://www.who.int/foodsafety/publications/mra_31/en/)
Specific comments
Title: An overview of Shiga-toxin producing Escherichia coli in Brazil: A
systematic review.
Can you just call it systematic review of Escherichia coli in Brazil?
Author Response
Paper Microorganisms - 484896
Response to Reviewer 1
We would like to thank for your considerations and thoughtful critique of our manuscript. We believe we have responded to all the concerns and suggestions, and thus improved the overall impact of review.
The modifications are in a red font in the text, and the questions are answered in this file.
Reviewer: This is a well written paper. There are a couple of recent publications that should be mentioned in the introduction as they are topical (see list below). The publications are not specific to Brazil do include the region. I think the authors need to be clear what this review is providing that is not in the others.
AU: Dear reviewer, we have added the mentioned papers and explained the highlight of our manuscript in contrast to other studies published in the literature: “There are some documents compiled global STEC data based on sites of health institutions and overview by continent [15,16]. However, Escherichia coli STEC serogroups spread in Brazil remain unclear. In this context, the aim of the present study was to conduct the first systematic review of Escherichia coli STEC with focus on Brazil and to compare the presence of serogroups detected in food products, animals and humans” (Line 68). Thank you for your suggestion.
Title: An overview of Shiga-toxin producing Escherichia coli in Brazil: A systematic review. Can you just call it systematic review of Escherichia coli in Brazil?
AU: Thanks for your suggestion, we have changed the title for: “Shiga-toxin producing Escherichia coli in Brazil: A systematic review”.

Reviewer 2 Report
Line 132: Change "whom" to "who had"
Line 148: Delete "world-wide". The word usage in the sentence is out of context.
Author Response
Paper Microorganisms - 484896
Response to Reviewer 2
We would like to thank for your considerations and thoughtful critique of our manuscript. We believe we have responded to all the suggestions.
The modifications are in red font in the text, and the questions are answered in this file.
Line 132: Change “whom” to “who had”
AU: Thank you for the suggestion, the word was altered as suggested by the reviewer (Line 134).
Line 148: Delete “world-wide”. The word usage in the sentence is out of context.
AU: Thank you for the suggestion, the word was deleted (Line 150).
